# An Easy-to-Use Prehospital Indicator to Determine the Severity of Suspected Heat-Related Illness: An Observational Study in the Tokyo Metropolitan Area

**DOI:** 10.3390/diagnostics13162683

**Published:** 2023-08-15

**Authors:** Junko Yamaguchi, Kosaku Kinoshita, Minami Takeyama

**Affiliations:** Division of Emergency and Critical Care Medicine, Department of Acute Medicine, Nihon University School of Medicine, Tokyo 173-8610, Japan; kinoshita.kosaku@nihon-u.ac.jp (K.K.);

**Keywords:** heat-related illness, heatstroke, prehospital care, prevention, severity

## Abstract

Rapid hospital arrival decreases mortality risk in heat-related illnesses. We investigated an easy-to-use indicator of life-threatening severity of heat-related illnesses in a community setting to enable quick hospitalization by using data extracted from prehospital transportation records of a database from 2016 that included information on the clinical severity of suspected heat-related illnesses in patients (*n* = 2528) upon hospital arrival. Patient-related risk factors (adjusted odds ratio, aOR [95% confidence interval, CI]) included age, vital signs, location of the patient, and illness severity, and respiratory rate (3.34 [1.80–6.22]), heart rate (2.88 [1.57–5.29]), axillary body temperature (7.79 [4.02–15.1]), and consciousness level (38.3 [5.22–281.1]) were independent risk factors for heat-related illness severity. On-site blood pressure was not an independent factor for illness severity. Heart rate > 120 beats/min, respiratory rate > 24 breaths/min, and temperature > 38.6 °C (highest areas under the receiver operating characteristic curves [95% CI]: 0.80 [0.75–0.87]; 0.73 [0.67–0.81]; and 0.83 [0.77–0.91], respectively) predicted life-threatening illness severity. Changes in the vital signs of patients with heat-related illnesses, particularly tachycardia and tachypnea, constitute sensitive, easy-to-use indicators that facilitate rapid identification of severity by laypersons and transport of patients before aggravation to a life-threatening situation.

## 1. Introduction

High summer temperatures, due to global warming, have emerged as an unavoidable cause of heat-related illness, and heat-related deaths have been reported during heat waves [1,2]. Heat-related illness is defined as a physiological insult to the body that occurs because of exposure to elevated temperatures, which results in the elevation of the core body temperature to levels that surpass the compensatory limits of thermoregulation [3]. Thus, heat-related illness comprises a set of syndromes that evolve along a continuum, which ranges from mild illness, such as heat cramps and heat exhaustion, to severe illness, such as fatal multiorgan failure with heatstroke [3,4].

Projection analysis for the next 100 years forecasts higher maximum temperatures, an increased number of hot days, and extreme climatic phenomena [5]. Moreover, an increase in health-related damage due to the increase in temperatures, both due to global warming and the induction of the “heat island phenomenon,” has been predicted [6]. In Japan, there has been a year-on-year increase in the number of patients with heat-related illnesses who are diagnosed by paramedical emergency medical technicians (EMT) in the community setting and then transported to the hospital in an ambulance. In the Tokyo metropolitan area, which is under the jurisdiction of the Tokyo Fire Department, the number of patients with heat-related illness has increased nearly threefold in a single decade, from 1041 patients in 2006 to 2885 patients in 2016 [7].

In particular, the mortality rate is high among patients with heatstroke who have impaired thermoregulation [8].

Heat-related illness in patients with pathological changes is aggravated by consecutive reactions, which may not be clearly demarcated. Therefore, initiating appropriate treatment before the onset of aggravated illness is necessary to improve the clinical outcome. Thus, with rapid recognition and diagnosis based on an effective clinical indicator for estimating the severity of heat-related illness, it is likely that most patients will recover quickly without major complications (e.g., central nervous system disturbances, impaired consciousness) and can be discharged. Many sports competitions, such as the 2020 Olympic Games in Tokyo, have been held in hot environments during the summer. Thus, in the community setting, it is very important to provide laypersons with tools to efficiently and effectively facilitate the identification of severe heat-related illnesses that could rapidly progress to a life-threatening condition. This study was conducted to clarify the characteristics and severity of heat-related illness, and then to identify a sensitive and easy-to-use indicator for utilization in the community setting. The key objectives of this study were to determine the incidence and severity of heat-related illness in the study population and identify specific criteria that facilitate early diagnosis at the scene.

## 2. Materials and Methods

This single-center observational study was conducted using prehospitalization data obtained from 1 January to 31 December 2016, from medical records in the database of the Tokyo Fire Department, which covers the Tokyo metropolitan area. Eligible participants were adult patients older than 20 years who were transferred to the hospital by paramedical EMTs in an ambulance in the Tokyo metropolitan area.

The prehospital transportation record had to include heat-related illnesses. In this study, heat-related illness was ascertained based on the diagnosis provided for approval of emergency transport of a patient. The diagnosis, which was coded as heat syncope, heat cramps, heat exhaustion, or heatstroke, was obtained from the Tokyo Fire Department database system.

Two coauthors with expertise in treating patients with heat-related illnesses reviewed the records in this database and identified 2968 cases during the study period with heat-related illness. Of the 2968 patients, 440 juvenile patients (≤19 years old) were excluded because of differences in the standard values of their vital signs compared to those of adults. Finally, this study included 2528 participants who were considered to have a heat-related illness (Figure 1).

The prehospital transportation record had to include heat-related illnesses. In this study, heat-related illness was based on the diagnosis provided for approval of emergency transport of a patient. The diagnosis, which was coded as heat syncope, heat cramps, heat exhaustion, or heatstroke, was obtained from the Tokyo Fire Department database system. The certification approval for emergency transport was issued by a doctor who noted the degree of illness severity (Grades 1–5) of the patient upon hospital arrival as follows: Grade 1, does not require hospitalization; Grade 2, necessitates hospitalization, but is not life-threatening; Grade 3, considered to be life-threatening; Grade 4, critical illness with impending danger of death; and Grade 5, death. 

The study population was divided into two groups based on the classification of illness severity as life-threatening or non-life-threatening. Grades 1 and 2, non-S group; Grades 3 and 4, S group; Grade 5 was not considered, as we excluded patients with out-of-hospital cardiopulmonary arrest.

The standard criteria for judging severe heat-related illness in the prehospital aid setting in Tokyo metropolitan area are as follows: (1) It occurs in a high-temperature environment. (2) Consciousness impairment is recognized. However, vital signs, which are included in the usual emergent setting and severity of illness at the prehospital stage, were not used as a guide for determining severe heat stroke. Patient-related risk factors, including age, vital signs and level of consciousness measured upon arrival at the site, the location of the patient, and the severity of the illness reported to gain approval for emergency transport, were extracted from the prehospital transportation records.

The certification approval for emergency transport was issued by a doctor who noted the degree of illness severity (Grades 1–5) of the patient upon hospital arrival as follows: Grade 1, does not require hospitalization; Grade 2, necessitates hospitalization, but is not life-threatening; Grade 3, considered to be life-threatening; Grade 4, critical illness with impending danger of death; and Grade 5, death. The participants were divided into two groups based on the severity of life-threatening conditions: the non-severe (non-S) group comprising patients with Grades 1 and 2, and the severe (S) group comprising patients with Grades 3 and 4. In an additional analysis, the participants were grouped differently to predict hospitalization requirements: the non-hospital (non-H) group comprising patients with Grade 1 and the hospital (H) group comprising patients with Grades 2–4. Grade 5 was not considered, as patients with out-of-hospital cardiopulmonary arrest were excluded. Furthermore, the classification of the level of consciousness was divided into two categories: alert or not alert. The JCS (Japan Coma Scale) score has been widely used to assess patients’ consciousness levels in the Japanese prehospital field [9]. JCS scores are divided into four main categories: alert (0), and one-digit, two-digit, and three-digit codes based on an eye response test, each of which has three categories. However, the GCS (Glasgow Coma Scale) [10] has not been used in prehospital care in Tokyo, Japan, and it was difficult to convert the JCS evaluation into the GCS because of the different evaluation methods. Therefore, we converted the level of consciousness of our participants from JCS to the Mayo Clinic Classification [11] (Appendix A). HR, RR, and BT measured in the axillae and the level of consciousness were measured by paramedics as soon as they arrived on scene.

The study did not enroll patients who received medical treatment at the hospital but were not transported by ambulance, or those who did not receive medical treatment at the hospital despite being transported to the hospital in an ambulance. Patients aged 70 years or older were classified as older patients in this study, and the study participants were divided into an older-adult group and a non-older-adult group [12].

### Statistical Analysis

Statistical analyses were performed using the JMP pro 13.2.1 statistical software package (SAS Institute, Cary, NC, USA), and the C-index and calibration curve were built by being computed on Rversion 4.3.1 (R Foundation for Statistical Computing) with the “rms” package and “calibration curve” package.

The concordance index (C-index) was computed on R with the “rms” package and the “Hmisc” package. [13]. Data are presented as the mean (standard deviation) or the number of cases (%). Continuous variables were compared using Student’s *t*-test or the Mann–Whitney *U* test, as appropriate. The chi-squared test was used to compare categorical variables. The clinical outcome (severe vs. non-severe, hospitalized vs. non-hospitalized) was predicted using multiple logistic regression through a stepwise increase in the variables and calculation of the odds ratios and 95% confidence intervals (CI). The stepwise increase in variables for analysis was applied to the previously described clinical factors that are related to the outcome (severe vs. non-severe, hospitalized vs. non-hospitalized) explanatory variables. All variables with a *p* < 0.2 in the bivariate model were included in the multivariate model and subjected to multiple logistic regression analysis [14]. To assess the ability of parameters to predict clinical severity, the diagnostic performance (sensitivity, specificity, negative and positive predictive values) of each variable was calculated based on the receiver operating characteristic (ROC) curves that were constructed, and the corresponding areas under the ROC curve (AUROC) were calculated to obtain cutoff values. Using ROC curve analysis, we determined the optimal cutoff points at a significance level of 5%. To perform the validation cohort of the obtained predictive model, the bootstrap method was used for internal validation. [15]. The 95% CI and AUCs were obtained by creating ROC curves with 1000 repetitions. The Hosmer–Lemeshow test was conducted to validate the optimization of the model obtained by multivariate logistic regression analysis of training data [16,17]. The C-index was calculated, and calibration plots were created to check whether the predicted results matched the actual distribution of the data [18].

## 3. Results

This study included 2528 patients considered to have a heat-related illness.

The subgrouping of patients, based on the degree of severity ascertained from the certificate of emergency transport, is shown in Table 1. There were 1484 patients with Grade 1 severity, 984 patients with Grade 2 severity, and 60 patients with Grade 3 or 4 severity. The average age of this study population was 62.8 years, and 1433 participants were older adults (56.7% of the total study population; Table 1 and Table 2). Of the total, 1483 cases of heat-related illness occurred indoors (Table 2). The incidence of each severity level and prevalence in each age group is shown in Figure 2.

The incidence of mild illness (Grade 1) was higher in the younger age group than in older adults. However, the incidence of severe illness, which required inpatient treatment and included all participants besides those with non-mild illness (Grades 2, 3, and 4), was higher in the older age groups (Figure 2a). A logistic regression analysis identified a significant decrease in the incidence rate of mild illness by age group (adjusted odds ratio (aOR), 0.976; 95% CI, 0.972–0.980, *p* < 0.001) and revealed that the incidence of heat-related illness was higher in the more advanced age groups (Figure 2b). The level of consciousness was divided into two categories: alert (1481 patients (58.6%)) or not alert (1047 patients (41.4%)). Univariate analyses were performed using clinical factors previously described related to heat-related illness severity. A comparative subgroup analysis of the non-S and S groups showed a statistically significant intergroup difference in severity (life-threatening or non-life-threatening) (Table 2).

Multiple logistic regression analyses were performed to identify the significant predictors of severity in patients with heat-related illness in prehospitalization conditions. Variables with *p*-values < 0.2 by bivariate analysis were then introduced into the multivariate model to take into account multicollinearity [14]. These analyses revealed that the respiratory rate (RR; aOR, 0.897; 95% CI, 0.855–0.941, *p* < 0.0001), heart rate (HR; aOR, 0.983; 95% CI, 0.968–0.998, *p* = 0.027), body temperature (BT; aOR, 0.509; 95% CI, 0.405–0.641, *p* < 0.0001), and consciousness level (aOR, 0.030; 95% CI, 0.004–0.225, *p* = 0.0001) at the prehospitalization stage were independently associated with illness severity in patients with heat-related illnesses. The blood pressure (BP) measured at the scene was not an independent factor for severity.

To assess the ability of the parameters HR, RR, and BT to predict clinical severity, the diagnostic performances (sensitivity, specificity, and negative and positive predictive values) of each variable were calculated. A ROC curve was constructed, and the corresponding AUROC was calculated. The ROC curve analysis showed that the values for HR, RR, and BT that distinguished the non-S and S groups of patients were 120 beats/min, 24 breaths/min, and 38.6 °C, respectively. The highest AUROCs (Figure 3) were observed for RR > 24 breaths/min at 0.73 (95% CI, 0.67–0.81), HR > 120/min at 0.80 (95% CI, 0.75–0.87), and BT > 38.6 °C at 0.83 (95% CI, 0.77–0.91).

The respective hazard ratios for the severity based on RR, HR, and BT values as prehospital triage criteria, which were dichotomized to ≥120 beats/min, ≥24 breaths/min, and ≥38.6 °C, respectively, (Table 3).

Another subgroup analysis of the non-H and H groups divided based on the hospitalization request showed statistically significant intergroup differences, except for the parameter diastolic pressure, which was similar to the differences between the non-S and S groups (Appendix A). However, the AUROC values for these variables were generally low. The highest AUROC values were observed for RR > 24 breaths/min at 0.54 (95% CI, 0.51–0.56), for HR > 70 beats/min at 0.60 (95% CI, 0.57–0.62), for BT > 37.1 °C at 0.52 (95% CI, 0.50–0.54), and for diastolic BP > 62 mmHg at 0.47 (95% CI, 0.44–0.49; Appendix A). Therefore, we did not perform a multiple regression analysis using these cutoff values as explanatory variables because they are unlikely to be useful criteria for ascertaining the need for hospitalization. The predictive model-derived equation for determining the severity of heat-related illness patients in prehospital rescue obtained from these results from multivariate logistic regression analysis (Appendix A) is as follows: predictive model equation = 7.79 × [BT > 38.6] + 3.34 × [RR > 24] + 2.88 × [HR > 120] + 0.026 × [Alert or Not Alert] + 61.1. The target rate was 97.7% if we used the model equation, and the highest areas under the receiver operating characteristic curves (AUROC) was 0.93, with a 95% confidence interval of 0.90–0.96 (Appendix A).

For performing of the validation cohort of our predictive model, bootstrapped c statistics [15] and calibration curves were used to assess external model discrimination and fit. We generated 1000 ROC (receiver operating characteristic) curves using the bootstrap method and calculated 95% confidence intervals for the AUC (area under the curve). We divided the groups into two groups: the training group and the test group [18].

We split the data 4:1 or 2:1 (random assignment) and obtained AUC values and 95% confidence intervals for the training and test groups, and the Hosmer–Lemeshow test was performed to validate the optimization of the model obtained by multivariate logistic regression analysis of the training data [16,17]. *p* > 0.05 was used for the cut-off value (Appendix A). As external validation, the concordance index (C-index) for the prediction nomogram was 0.93 for the test group in the case of splitting the patient data of our study randomly by 4:1 (training: test = 4:1), and the C-index for the prediction nomogram via bootstrapping validation was 0.93 for the training group. As external validation, the C-index for the prediction nomogram was 0.95 for the test group in the case of splitting the patient data of our study randomly by 2:1 (training: test = 2:1), and the C-index for the prediction nomogram was 0.92 for the training group. Consequently, the prediction model had the same discriminant power in the training cohort as in the test group cohort (Appendix A).

As shown above, the accuracy of the prediction model equation is high. However, the results might overestimate the accuracy of the prediction equation because of existing multicollinearity. Multicollinearity is assessed using variance inflation factors (VIF) [19]. The potential multicollinearity was found to be present in RR, HR, and BT (VIF = 1.041, 1.138, and 1.127, respectively), and the number of only sixty severe cases of our study was very small. In addition, it might be difficult to use the prehospital stage of the complexity of the prediction model equation itself.

## 4. Discussion

This study was conducted to, firstly, identify factors related to the clinical severity of patients with heat-related illness in the prehospital setting and, secondly, to identify an easy-to-use prehospital indicator of the clinical severity of patients with heat-related illness. This indicator must be easy to use by ordinary citizens to facilitate early recognition of the severity of heat-related illness. This severity indicator is useful for caregivers because the older-adult group in this study was at a greater risk for more severe heat-related illness. Data analysis showed that the optimal cutoff points for HR, RR, and BT were 120 beats/min, 24 breaths/min, and 38.6 °C, respectively, for use as a prehospital indicator to assess the illness severity of patients with heat-related illness. Interestingly, BP did not indicate the severity of illness in the prehospital setting. Easier indicators for recognition of severity in patients with heat-related illness for caregivers in community settings would be preferable. Thus, these factors can be utilized by laypersons while focusing not only on the high BT but also tachypnea and tachycardia in patients with suspected heat-related illness, in order to prioritize the transportation of patients to designated emergency care centers based on their condition.

Higher temperatures in summer due to global warming create an increase in heat-related illnesses [1,2]. In an observational study, Hausfater et al. [20] aimed to determine independent mortality factors for heatstroke in a study population of 1456 patients. The patients suffered from heat-related illnesses and core temperatures of >38.5 °C were measured upon arrival in the emergency room at 16 medical facilities. Similarly, Davido et al. [21] undertook a single-center study to identify early phase independent predictive factors of mortality risk in 165 patients with heat-related illness, core temperatures of >38.0 °C, and dehydration. In addition, Misset et al. [2] considered independent risk factors associated with higher mortality risk in 345 patients with heat-related illness from 80 study centers. The findings from these abovementioned three studies suggested that the patient’s vital signs upon hospital arrival were related to poor outcomes of heat-related illness.

An international classification for the severity of heat-related illness was applied in this study, which includes heat cramps and syncope, heat exhaustion, and heatstroke, and it has been used globally to assess the severity of heatstroke based on the patient’s symptoms and pathophysiology [8,21]. The abovementioned international classification has a wide application due to its ease of use and sensitivity in the identification of patients with heat-related illnesses who are at risk of fatal multiorgan failure. This international classification characterizes severe heat-related illness using elevated core temperature and central nervous abnormalities as definitive criteria [8].

Impaired consciousness and high axillary temperature were also independent factors of heat-related illness severity in our study. However, an axillary temperature reading might be more susceptible to changes in response to external factors before the patients are transferred to the hospital than other measures of core temperature (rectal or tympanic). In the prehospital or community setting, only paramedics or other medical staff may obtain an accurate core temperature measurement. Therefore, it might be difficult for medical staff to precisely determine the clinical severity of the patient using only the international classification. The novel indicators of the severity of heat-related illness in prehospital settings described in this study may improve the outcome of patients as they do not rely on measurements of the core body temperature. This could decrease the likelihood of severe cases being underestimated by medical staff, paramedics, or even the general public, including caregivers. The indicators should be focused on early recognition of heat-related illness. Overestimated triage regarding the recognition of heat-related illness is permissible because it will initiate rapid treatment that could improve the prognosis of heat-related illness. Thus, there is a need for the utilization of the proposed, more sensitive indicators to enable recognition of any deterioration of the clinical status of patients with heat-related illnesses in the prehospitalization setting. Well-designed prospective studies are needed to clarify the cutoff points for abnormalities in physical signs, such as consciousness or vital signs, to avoid progressive worsening of the patient’s clinical condition.

Previous reports have suggested that altered vital signs upon admission are related to poor patient outcomes [2,21]. However, BP was unrelated to the patient’s illness severity in our study. The differences in our results compared to those of previous reports might be attributed to compensatory mechanisms that affect the heart rate. In addition, the patient’s temperature was determined to be a dependent prehospital triage criterion, as other variables often influence the temperature in the prehospital setting. Thus, there is a need for greater attention to changes in vital signs, especially regarding tachycardia and tachypnea, to prevent accelerated worsening of the clinical condition in the prehospitalization setting. As abnormal RR and HR values are easily recognizable by both healthcare providers and laypersons, these criteria can be useful to recognize early signs of illness severity in patients with heat-related illnesses before its evolution into a life-threatening condition.

### Limitations

This study has several limitations. First, it was a retrospective observational study conducted using data derived during prehospital care in Tokyo, which limited the number of factors that were available for analysis. The criteria by which an EMT or layperson in the field determined that a case was a heat-related illness were not rigorous, and it was impossible to obtain biochemical test or imaging data. The inability to regulate BT is a continuum that has symptoms and severity that range from heat syncope to organ failure to life-threatening heatstroke [22]. Moreover, heat-related illness can have a wide range of causes, from exertion- or drug-induced hyperthermia to environmental hyperthermia, known as heat stroke. The definitive diagnosis of heat-related illness requires the exclusion of various differential diagnoses (sepsis, thyroid dysfunction, malignant syndrome, serotonin syndrome, drug withdrawal symptoms, etc.) [23]. Furthermore, both definitive diagnosis and severity assessment require detailed patient information; examination at a hospital, including evaluation of medical history and medications; and various blood tests and imaging studies at the hospital. The available data did not include such detailed patient information in prehospital care, and the severity was subjectively assessed by a physician when the patient arrived at the hospital, before blood samples were collected or imaging tests were performed. Therefore, it is unclear whether the included cases were strictly heat-related illnesses. Second, the clinical characteristics of patients with heat-related illnesses in the Tokyo metropolitan area could not be captured comprehensively because the study population did not include patients who received medical treatment in an ambulance and those who did not receive medical treatment at the hospital despite being transported in an ambulance. Third, the characteristics of non-heat-related illness could not be investigated simultaneously and in the same area. Fourth, other factors that might alter vital signs were not assessed in this study; however, the effects of preexisting diseases (heart failure, chronic obstructive pulmonary disease, diabetes mellitus, infectious diseases, skin diseases, sickle cell disease, etc.), risk factors for heat-related illnesses (age, obesity, dehydration, alcohol consumption, etc.), and the use of medication (e.g., anticholinergics, diuretics, anticholinergics, beta-blockers, Ca-blockers, antihistamines, amphetamines, etc.) should be considered [24,25,26]. Finally, data from prehospital emergency settings have not been analyzed, and we have not been able to identify the severity of heat-related illness. The heart rate and respiratory rate are predicted to be confounders for body temperature. Confounding is also predicted by age and sex, as well as underlying illnesses. In fact, it was confirmed that there is a collinearity between respiratory rate and body temperature of our study. We analyzed the most appropriate method to eliminate confounding. However, we could not remove completely exclude the confounders mathematically. Therefore, the results of the severity-related factors obtained by our method are affected by confounders, and the obtained *p*-values might be overestimated. However, our study aimed not primarily at clinicians in a hospital setting, but to evaluate whether triage criteria can be simplified for the handling of patients with suspected heat-related illness by laypersons in the field without the need for advanced medical response skills.

In future studies, the relationship between a patient’s vital signs obtained in prehospital settings and the objective severity assessment as judged in a medical institution should be investigated in patients with suspected heat-related illness. This might facilitate the development of useful triage criteria that EMTs and laypersons can use to determine the need for hospitalization and to assess the illness severity of patients with suspected heat-related illness.

## 5. Conclusions

In the prehospital phase, besides impaired consciousness, which has traditionally been considered as a severity indicator for heat-related illness, an RR > 24 breaths/min, HR > 120 beats/min, and BT > 38.6 °C, but not BP, are potentially sensitive indicators for a severe heat-related illness suspected at the prehospital stage. Thus, attention should be focused on changes in vital signs, especially tachycardia and tachypnea, in the prehospitalization stage. Both healthcare providers and laypersons can easily determine these two parameters. This easy-to-use indicator that incorporates the HR and RR can facilitate early recognition of the clinical severity of heat-related illness in patients before their progression to a life-threatening situation.

## Figures and Tables

**Figure 1 diagnostics-13-02683-f001:**
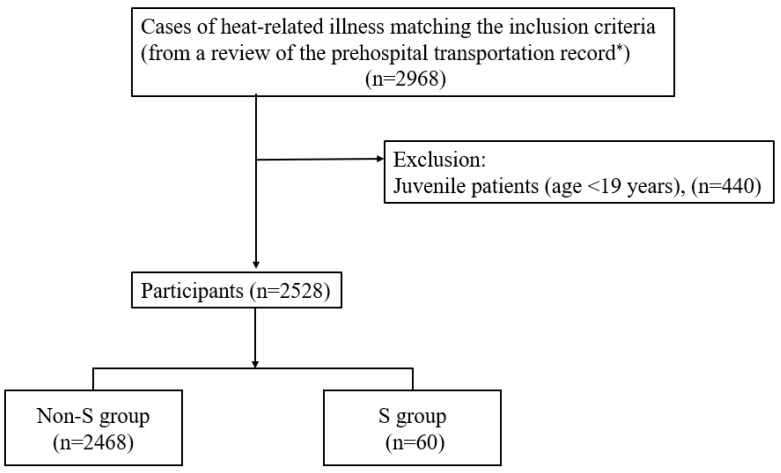
Flow chart the inclusion process of the study. * Eligible participants were adult patients older than 20 years who were transferred to the hospital by paramedical EMTs in an ambulance in the Tokyo metropolitan area.

**Figure 2 diagnostics-13-02683-f002:**
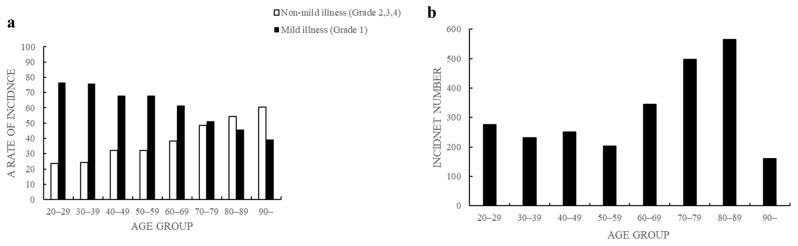
Incidence of heat-related illness, degree of illness severity, and number of cases by age group. (**a**) Incidence of heat-related illness by degree of illness severity in each age group. A logistic regression analysis identified a significant decrease in the incidence rate of mild illness by age group (adjusted odds ratio (aOR), 0.976; 95% confidence interval (CI), 0.972–0.980, *p* < 0.001). (**b**) Incidence (number of cases) of heat-related illness in each age group.

**Figure 3 diagnostics-13-02683-f003:**
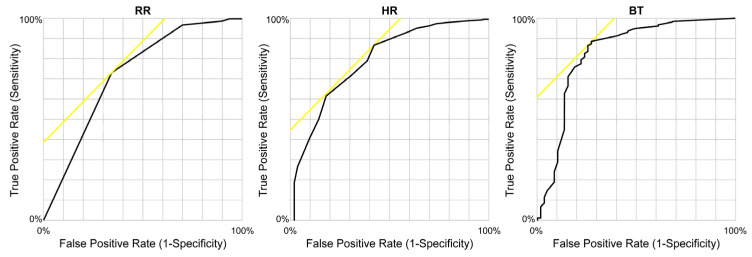
Receiver operating characteristic curves for severity in patients with heat-related illness in the prehospitalization setting. The ROC curve analysis showed that the optimal cutoff values for HR, RR, and BT were 120/min, 24/min, and 38.6 °C, respectively. The highest AUROC values were found for RR > 24/min with 0.73 (95% CI, 0.67–0.81), HR > 120/min with 0.80 (95% CI, 0.75–0.87), and BT > 38.6 °C with 0.83 (95% CI, 0.77–0.91). Abbreviations: AUROC, area under the ROC curve; BT, body temperature (axillary temperature); CI, confidence interval; HR, heart rate; ROC, receiver operating characteristic; RR, respiratory rate.

**Table 1 diagnostics-13-02683-t001:** Classification of severity (Grades 1–5) assigned in the certificate of emergency transport ^1^.

Grade	Non-Older-Adult	Older Adult ^2^	Total (%)
1	775	705	1484 (58.7%)
2	302	682	984 (38.9%)
3	13	36	49 (1.9%)
4	5	6	11 (0.4%)
5	0	0	0 (0%)
Total	1095	1433 (56.7%)	2528

^1^ The certificate authorizing emergency transport is provided by a doctor when the patient arrives at the hospital in an ambulance; the degree of clinical severity upon admission of the patient is documented in this certificate as follows: Grade 1, does not require hospitalization; Grade 2, necessitates hospitalization, but is not life-threatening; Grade 3, considered to be life-threatening; Grade 4, critical illness with impending danger of death; and Grade 5, death or a patient who is in cardiopulmonary arrest on hospital admission. ^2^ Patients older than 70 years were classified as older patients in this study.

**Table 2 diagnostics-13-02683-t002:** Intergroup differences in patient characteristics (prehospital setting) ascertained using univariate analysis ^1^.

Characteristic	All (*n* = 2528)	Non-S Group (*n* = 2468)	S Group (*n* = 60)	*p*-Value ^2^
Age (years)	62.8 ± 21.7	62.6 ± 0.44	69.9 ± 2.61	0.006
Sex (M/F) [M/total (%)]	1532:996 [60.6]	1496:972 [60.6]	36:24 [60.0]	0.88
Systolic BP (mmHg)	126.6 ± 31.6	126.7 ± 25.3	124.4 ± 38.9	0.66
Diastolic BP (mmHg)	71.2 ± 15.2	71.2 ± 15.2	71.2 ± 18.4	0.95
Heart rate (bpm)	95 ± 20.8	94 ± 20.3	120.4 ± 25.8	<0.001
Respiratory rate (bpm)	20 ± 3.92	19.9 ± 3.67	25.4 ± 7.92	<0.001
Body temperature (°C)	37.1 ± 1.2	37.1 ± 1.1	39.2 ± 1.62	<0.001
Location (indoor:outdoor) [indoor/total (%)]	1483:1045 [58.7]	1447:1021 [58.6]	36:24 [60.0]	0.87
Older adults (older adults:non-older adults) [older/total (%)]	1433:1095 [56.7]	1391:1077 [56.4]	42:18 [70.0]	0.042
Consciousness level (alert:not alert) [alert/total (%)]	1481:1047 [58.6]	1480:988 [60.0]	1:59 [1.67]	<0.0001

Data are presented as the mean ± standard deviation, or frequency (proportion). ^1^ The study population was divided into two groups based on the classification of illness severity as life-threatening or non-life-threatening. Grades 1 and 2, non-S group; Grades 3 and 4, S group; Grade 5 was not considered, as we excluded patients with out-of-hospital cardiopulmonary arrest. Axillary body temperature was measured. Patients older than 70 years were classified as older patients in this study. The classification of the level of consciousness was divided into two categories: alert or not alert. ^2^ The Mann–Whitney *U* test and the chi-squared test were performed. Abbreviations: BP, blood pressure; M/F, male/female; non-S, non-severe; S, severe.

**Table 3 diagnostics-13-02683-t003:** Multivariate logistic regression analysis of factors associated with the severity of heat-related illness (prehospitalization setting).

Factor	Correlation Coefficient	SE	*p*-Value	Odds Ratio	95% CI
RR > 24/min	1.207	0.317	<0.0001	3.34	1.80–6.22
HR > 120/min	1.057	0.310	0.001	2.88	1.57–5.29
BT > 38.6 °C	2.053	0.338	<0.0001	7.79	4.02–15.1
Alert or not	3.646	1.017	<0.0001	38.3	5.22–281.1

Multivariate logistic regression analyses (stepwise increase in the number of variables) were conducted with the explanatory variables older adults, respiratory rate, heart rate, temperature, and consciousness level. Abbreviations: BT, body temperature (axillary temperature); CI, confidence interval; HR, heart rate; RR, respiratory rate; SE, standard error. Patients older than 70 years were classified as older adults in this study.

## Data Availability

Our research has been approved by the Tokyo Fire Department. The data generated and analyzed during the current study are not publicly available because only officers of the Tokyo Fire Department can access the database. One of the authors, M.T., accessed the database and extracted these data for our research. Derived data, without patient’s personalized information, that support the findings of this study are available from the corresponding author, J.Y., on reasonable request.

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
