# Peer review of "An Easy-to-Use Prehospital Indicator to Determine the Severity of Suspected Heat-Related Illness: An Observational Study in the Tokyo Metropolitan Area"

_diagnostics, 2023, doi:10.3390/diagnostics13162683_

Round 1

Reviewer 1 Report

Thank you for giving me the opportunity to review the article titled "An Easy-to-use Prehospital Indicator for Assessing the Severity of Suspected Heat-related Illness." Although the article was well-written in English, it did not present substantial or valuable information. Merely relying on routine vital signs such as respiratory rate (RR), heart rate (HR), body temperature (BT), and level of consciousness to determine the severity of heat-related illness would only create confusion in a clinical setting. To address this concern, it would have been beneficial for the author to employ predictive regression models with multiple variables for the final outcome. Additionally, it is important to provide the positive prediction value and negative prediction value in order to evaluate the accuracy of the predictions. Furthermore, the article lacks information regarding the calibration curve for prediction. It would also be helpful to know which region of the body was used for measuring body temperature. Moreover, considering the dynamic nature of RR, HR, and BT data, it is crucial to acknowledge the significance of the measurement point and the duration of exposure to the measurement, which unfortunately were omitted.

Author Response

We thank you for the insightful comments. We have attached a word document with our response to the comments.
We would like you to refer it.

Reviewer 2 Report

As the authors point out in their introduction, we're inevitably going to have to confront rising temperatures all over the world, and thus an increase in the incidence of heatstroke. Therefore, predicting the risk of progression to a severe form at an early stage, before admission to the hospital, like the quick SOFA for septic shock, can be a valuable guide for adapting care.

I have 2 important and related comments about this study:

The first is how is patient severity assessed? In fact, if the same parameters are used to assess severity (grade) and predict severity, it is not very surprising to obtain a predictive character... Even if the assessment is made by 2 different people, the parameters are the same. In other words, if heart rate, respiratory rate and blood pressure are used to classify patients into grades of severity, it is statistically expected that these parameters will predict severity...

My second comment is why didn't the authors use an independent validation cohort? This would allow predictive factors to be established on a first cohort and independently validated on a second one...

I also have a few minor comments:

Page 2, lines 46-50 and 61-65, which are repetitive, should be reworded in the introduction.

Page 2, lines 79-83 : A flow chart would be an elegant addition to the first part of the Methods chapter.

Page 3, line 100 : How are alert and non-alert patients separated? What is the Glasgow score value?

Page 4, table 2 : Could the authors comment on the fact that the patients' average temperature was 37.1°C?

Page 5, table 3 : How did the authors take into account the possible collinearity of variables such as heart rate and temperature?

Author Response

We have attached a word document with our response to the comments.
We would like you to refer it.

Round 2

Reviewer 1 Report

The revision had improved a lot and I had no further comments. Thank you.

Reviewer 2 Report

First of all, I'd like to thank the authors for answering all my questions.

These answers and the corrections made have helped to clarify my understanding and, I hope, to improve the manuscript.

For me, there is still one "major" methodological pitfall: the absence of a validation cohort. In fact, the same data are used to carry out the multivariate analysis (logistic regression) and to validate its results (ROC curves). So, if the results of the multivariate analysis are satisfactory, there's no reason why the « validation ROC curves » should be any different.

Thank you also for your answers concerning the collinearity of variables. However, you do not present either the correlation coefficient or the Variance Inflation Factor for the potential collinearity between Heart Rate and Respiratory Rate, which are part of the main elements of the results.

Thanks also for the flow chart. Perhaps including the number of patients hospitalized (1044) and the number of severely ill patients (60) could also clarify the results.
